# The Effect of Diabetes on Prognosis Following Myocardial Infarction Treated with Primary Angioplasty and Potent Antiplatelet Therapy

**DOI:** 10.3390/jcm9082555

**Published:** 2020-08-06

**Authors:** Stanislav Simek, Zuzana Motovska, Ota Hlinomaz, Petr Kala, Milan Hromadka, Jiri Knot, Ivo Varvarovsky, Jaroslav Dusek, Richard Rokyta, Frantisek Tousek, Michal Svoboda, Alexandra Vodzinska, Jan Mrozek, Jiri Jarkovsky

**Affiliations:** 1Second Department of Medicine—Department of Cardiovascular Medicine, First Faculty of Medicine, Charles University and General University Hospital, 12808 Prague, Czech Republic; ssime@lf1.cuni.cz; 2Cardiocenter, Third Faculty of Medicine, Charles University and University Hospital Kralovske Vinohrady, 10034 Prague, Czech Republic; knot@centrum.cz; 3First Department of Internal Medicine—Cardioangiology, ICRC, Faculty of Medicine of Masaryk University and St. Anne’s University Hospital, 65691 Brno, Czech Republic; ota.hlinomaz@fnusa.cz; 4Department of Internal Medicine and Cardiology, Faculty of Medicine of Masaryk University and University Hospital, 65991 Brno, Czech Republic; pkala@fnbrno.cz; 5Department of Cardiology, University Hospital and Faculty of Medicine Charles University, 30460 Plzen, Czech Republic; HROMADKA@fnplzen.cz (M.H.); ROKYTA@fnplzen.cz (R.R.); 6Cardiology Centre AGEL, 53003 Pardubice, Czech Republic; ivovarvarovsky@gmail.com; 7First Department of Internal Medicine, University Hospital and Faculty of Medicine Charles University, 50005 Hradec Kralove, Czech Republic; jaroslav.dusek@fnhk.cz; 8Cardiocenter—Department of Cardiology, Regional Hospital, 37001 Ceske Budejovice, Czech Republic; tousek@nemcb.cz (F.T.); jarkovsky@iba.muni.cz (J.J.); 9Institute of Biostatistics and Analyses at the Faculty of Medicine and the Faculty of Science of Masaryk University, 62500 Brno, Czech Republic; svoboda@iba.muni.cz; 10AGEL Research and Training Institute—Trinec Branch, Cardiovascular Center, Podlesi Hospital, 73961 Trinec, Czech Republic; avodzinska@centrum.cz; 11Cardiovascular Department, University Hospital Ostrava, 70800 Ostrava, Czech Republic; honzamrozek@email.cz

**Keywords:** acute myocardial infarction, primary percutaneous coronary intervention, diabetes mellitus, prognosis, antiplatelets, prasugrel, ticagrelor, clopidogrel

## Abstract

Purpose: To investigate the prognostic significance of diabetes mellitus (DM) in patients with high risk acute myocardial infarction (AMI) treated with primary percutaneous coronary intervention (pPCI) in the era of potent antithrombotics. Methods: Data from 1230 ST-segment elevation myocardial infarction (STEMI) patients enrolled in the PRAGUE-18 (prasugrel vs. ticagrelor in pPCI) study were analyzed. Ischemic and bleeding event rates were calculated for patients with and without diabetes. The independent impact of diabetes on outcomes was evaluated after adjustment for outcome predictors. Results: The prevalence of DM was 20% (N = 250). Diabetics were older and more often female. They were more likely to have hypertension, hyperlipoproteinemia, multivessel coronary disease and left main disease, and be obese. The primary net-clinical endpoint (EP) containing death, spontaneous nonfatal MI, stroke, severe bleeding, and revascularization at day 7 occurred in 6.1% of patients with, and in 3.5% of patients without DM (HR 1.8; 95% CI 0.978–3.315; *p* = 0.055). At one year, the key secondary endpoint defined as cardiovascular death, spontaneous MI, or stroke occurred in 8.8% with, and 5.5% without DM (HR 1.621; 95% CI 0.987–2.661; *p* = 0.054). In those with DM the risk of total one-year mortality (6.8% vs. 3.9% (HR 1.773; 95% CI 1.001–3.141; *p* = 0.047)) and the risk of nonfatal reinfarction (4.8% vs. 2.2% (HR 2.177; 95% CI 1.077–4.398; *p* = 0.026)) were significantly higher compared to in those without DM. There was no risk of major bleeding associated with DM (HR 0.861; 95% CI 0.554–1.339; *p* = 0.506). In the multivariate analysis, diabetes was independently associated with the one-year risk of reinfarction (HR 2.176; 95% Confidence Interval, 1.055–4.489; *p* = 0.035). Conclusion: Despite best practices STEMI treatment, diabetes is still associated with significantly worse prognoses, which highlights the importance of further improvements in the management of this high-risk population.

## 1. Introduction

Patients with diabetes mellitus (DM) and acute myocardial infarction (AMI) are at high risk for recurrent cardiovascular events [1,2], in part due to a greater tendency towards thrombosis [3,4]. Diabetics are characterized by increased platelet reactivity [5] including higher reactivity while on antiplatelet treatment [6,7]. ST-segment elevation myocardial infarction (STEMI) is characterized by a highly prothrombotic state [8], the highest of which can be observed in diabetic STEMI patients [9].

Primary percutaneous coronary intervention (pPCI) is the most effective and most recommended therapeutic approach in patients with STEMI and in those at very high-risk myocardial infarction without persistent ST-segment elevation (non-STEMI with ongoing ischemia) [10].

Prasugrel and ticagrelor are the currently recommended treatment in patients with acute coronary syndromes, including STEMI, since they have been shown to reduce ischemic events compared to clopidogrel [11].

Studies testing newer P2Y_12_ inhibitors in acute coronary syndromes showed similar positive effects for ticagrelor and prasugrel on outcomes of patients with DM compared to non-diabetics [1,12]. However, the AMI populations in these studies were heterogeneous and included minor infarcts, in which the AMI diagnosis was based on highly sensitive tests for troponins. As such, the subgroup analyses of these studies provided only limited information on high-risk AMI populations treated with primary or immediate PCI.

The effect of prasugrel and ticagrelor, specifically on the STEMI population, has been even less well studied. To date, only the ATLANTIC trial selectively tested novel P2Y_12_ inhibitors in a STEMI population. The independent effect of diabetes on prognoses in the highest risk patients following an AMI, treated with improved, up-to-date techniques, is therefore still open to discussion.

The multicenter randomized PRAGUE-18 study was a comparison of prasugrel and ticagrelor in patients with AMI indicated to primary PCI [13,14]. The trial was unique since the AMI population was real life (with very few exclusion criteria) and homogenous with respect to the highest thrombotic risk.

This sub-study aims to evaluate the prognostic significance of DM in patients with AMI treated with pPCI in the era of potent antithrombotics and to investigate whether the most efficient treatment currently available, i.e., primary PCI and potent P2Y12 inhibitors can change the negative impact of DM on the prognoses of patients at the highest risk of major adverse cardiovascular events.

## 2. Methods

This analysis includes subjects randomized into the PRAGUE-18 trial stratified by DM status and by insulin treatment into prespecified subgroups. Subjects were classified according to the presence or absence of DM at baseline. The diagnosis of DM was based on patients’ history and on initial clinical examination. All diabetics on a diet, oral hypoglycemic medication, or insulin control were included. A subgroup of patients requiring insulin control was evaluated separately as these patients present mostly more advanced DM and are at higher risk. Nevertheless, the reasons for pre-randomization choice between insulin and other treatments were not analyzed. Potential risk factors in patient histories, pre-, peri-, and post-procedural pharmacotherapy, and characteristics related to the pPCI procedure were searched and analyzed for subgroup differences. The impact of DM and the impact of insulin treatment, relative to glucose control, on patient prognosis (study endpoints) were evaluated separately using multivariate analyses.

Enrollment criteria, the design, and the randomization process of the PRAGUE-18 study have been previously described [13,14]. Briefly, P2Y_12_ inhibitor naive patients with AMI (STEMI and very high-risk non-STEMI) indicated to pPCI were randomized either to a prasugrel or ticagrelor loading dose and one-year therapy on top of aspirin treatment. Simple randomization using GraphPad scientific software was used in the study. The “sealed envelope” method was used for the distribution of the randomization codes. Since expenses for both drugs were not covered by insurance, patients were allowed to switch to clopidogrel during the study, under supervision of the treating physician. The study population consists of 1230 patients enrolled between May 2016 and November 2017. Hemodynamic instability was not an exclusion criterion for study participation. Nearly 4% of patients were in cardiogenic shock at baseline, and 5.2% were on mechanical ventilation. Almost all patients (99.2%) enrolled in the study underwent immediate PCI; primary PCI was performed in 94.6%. Radial access was used in two-thirds of patients and at least one intracoronary stent was implanted in 96% of patients.

The primary net-clinical endpoint was death, spontaneous MI, stroke, severe bleeding, or revascularization within 7 days. The secondary key efficacy endpoint was cardiovascular death, spontaneous MI, and stroke at 30 days and one year.

The occurrence of secondary endpoints, i.e., all-cause death, definite stent thrombosis (according to the Academic Research Consortium criteria), and bleeding (defined according to TIMI (Thrombolysis in Myocardial Infarction) and Bleeding Academic Research Consortium criteria) were also recorded. Data from the study were recorded using web-based case report forms and stored in a database system. An endpoint adjudication committee verified all study endpoints [13].

The study design was approved by the multicenter ethics committee at the University Hospital Kralovske Vinohrady in Prague, Czech Republic (EK-VP/04/2013), and by the ethics committees of all participating sites. Study protocol is registered under PRAGUE-18 Clinicaltrials.gov NCT02808767.

### Statistical Analysis

Continuous variables are presented as means or medians, and categorical variables are presented as numbers and percentages. The statistical significance of differences in categorical variables between patient groups was tested using the Fisher exact test; the Mann–Whitney *U* test was used for continuous variables. The occurrence of events over time was described and visualized using the Kaplan–Meier methodology; the statistical significance of differences between groups was tested using the log-rank test. The 1-dimensional and multidimensional Cox proportional hazards model were used for endpoint adjudications and described using hazard ratios (HR), 95% confidence intervals (CI), and statistical significance. In the multivariate analyses, the impact of DM on the prognosis was adjusted for age, sex, and presence of multivessel disease. All analyses were performed using SPSS version 24.0.0.1 (IBM Corporation, Armonk, NY, USA).

## 3. Results

Diabetic patients. DM was present in 250 (20%) patients. Prior to randomization, one-fourth (N = 64) of the diabetic patients were on long-term insulin treatment, 58.4% on oral hypoglycemic medication, and 16% were controlled by the diet only.

Baseline clinical characteristics of patients with and without DM, including initial ECG changes, chronic therapy before admission, initial laboratory results, and characteristics related to index angiography, are summarized in Table 1.

Diabetic patients were more likely to be older, female, obese, with hypertension, hyperlipidemia, and a history of bleeding. DM was also associated with a more frequent presence of multivessel coronary disease and left main disease. In patients on insulin, the presence of multivessel coronary disease and left main disease was especially high (64% compared to 6.3% for non-diabetics).

The median time from symptom onset to hospital arrival was significantly longer in diabetic patients on insulin therapy compared to patients without DM (3.5 vs. 2.5 h; *p* = 0.030).

Before randomization, subjects with DM were more likely to receive chronic treatment with aspirin, angiotensin-converting enzyme inhibitors, angiotensin receptor blockers, beta-blockers, and statins.

Drug-eluting stents were used in two-thirds and thrombo-aspiration during pPCI was used in one-third of patients with and without DM. Glycoprotein IIb/IIIa inhibitors were used more often in patients with DM (Table 1). The most frequent use of glycoprotein IIb/IIIa inhibitors was indeed observed in patients treated with insulin (28.1%). The result of pPCI in the DM group was more frequently evaluated by the treating interventional cardiologist as suboptimal or a procedural failure with the highest rate being patients on insulin (14.3%).

Discharge medications included aspirin in 97.3%, β-Blockers in 81.9%, ACE inhibitors/ARBs in 83.7%, and statins in 93.7% of the study population and did not differ between patients with and without DM (*p* = 0.999, *p* = 0.134, *p* = 0.626 and *p* = 0.270 respectively). The percentage of patients who switched their study treatment during the 12-month study course to clopidogrel (40.8%) did not differ between patients with or without DM (38.7%), *p* = 0.562.

Outcomes: The primary net-clinical endpoint occurred in 6.1% of patients with, and in 3.5% of patients without DM (HR 1.8; 95% CI 0.978–3.315; *p* = 0.055) (Figure 1A). At one year, the key secondary endpoint (CV death, spontaneous MI, and stroke) occurred in 8.8% of diabetics and in 5.5% of patients without DM (HR 1.621; 95% CI 0.987–2.661; *p* = 0.054) (Figure 1B). The total one-year mortality and risk of nonfatal reinfarction in DM vs. non-DM were 6.8% vs. 3.9% (HR 1.773; 95% CI 1.001–3.141; *p* = 0.047) and 4.8% vs. 2.2% (HR 2.177; 95% CI 1.077–4.398; *p* = 0.026) (Figure 1C,D), respectively; cardiac mortality at one year was 4% vs. 3% (HR 1.366; 95% CI (0.666–2.804; *p* = 0.393), respectively. The risk of definite stent thrombosis was higher in diabetics (HR 2.37; 95% CI 0.864–6.541; *p* = 0.08) (Appendix A). There was no risk of major bleeding related to the presence of DM (HR 0.861; 95% CI 0.554–1.339; *p* = 0.506) (Appendix A).

Patients with more advanced DM requiring insulin control exhibited all assessed endpoints more frequently compared with diabetic patients, not on insulin. The occurrence of the primary net clinical endpoint in this group of patients was 10.9% (HR 3.359; 95% CI 1.486–7.594; *p* = 0.002 in comparison to the reference group of patients without DM) (Figure 2A), the occurrence of the combined key ischemic endpoint at one year was 12.5% (HR 2.408; 95% CI 1.146–5.059; *p* = 0.017) (Figure 2B). The total mortality and risk of reinfarction at one year in DM patients on insulin was 12.5% (HR 3.343; 95% CI 1.559–7.165; *p* = 0.001) (Figure 2C) compared to 6.3% for patients without DM (HR 2.979; 95% CI 1.027–8.646; *p* = 0.036) (Figure 2D).

Using the Cox proportional hazards multivariate model adjusted for significant predictors of prognosis (see Methods section), DM remained an independent predictor of recurrent AMI at one year (HR 2.176; 95% CI 1.055–4.489; *p* = 0.035). Presence of DM on insulin was an independent predictor of total one-year mortality (HR 2.642; 95% CI, 1.223–5.709; *p* = 0.013).

## 4. Discussion

### 4.1. Study Population

The impact of DM on outcomes of patients with AMI has been explored in several previous studies. In contrast, this analysis examined a very precisely defined population of patients with the highest risk AMI. All enrolled patients received best practices treatment, i.e., pPCI using new-generation drug-eluting stents plus, new, potent oral antiplatelet therapy. Furthermore, the penetration of secondary preventive medication at discharge was very high in the PRAGUE-18 study population, including aspirin, statins, beta-blockers, and ACE inhibitors/ARB inhibitors.

### 4.2. Antiplatelet Treatment

All patients received potent antiplatelet therapy consisting of aspirin and ticagrelor or prasugrel. These newer (ticagrelor and prasugrel) antiplatelet agents are believed to overcome the problem with platelet resistance to clopidogrel, and they reduce thrombotic complications to a greater extent than clopidogrel in DM patients [1,12]. Data shows that antiplatelet therapy provided with prasugrel is of particular benefit to patients with DM. Subjects with DM had a greater reduction in ischemic events without an increase in major bleeding and therefore had a greater net treatment benefit with prasugrel compared to patients without DM [12]. Ticagrelor exerts similar or greater inhibition of platelet reactivity compared to prasugrel in DM patients with coronary artery disease [15]. Therefore, we hypothesized that we would observe a less negative impact of DM on the prognosis of patients in PRAGUE-18 compared to older studies.

### 4.3. Risk Factors

Similar to other published cohorts of AMI patients [1,16], DM patients were more likely to have several risk criteria, including multivessel disease, compared to non-diabetic patients. In some pPCI studies, longer time from symptom onset to hospital arrival was observed with DM patients [16,17]. In our study, this time difference was significant only with regard to DM patients treated with insulin, who, on average, arrived at the hospital one hour later than non-diabetics. This reflects frequent atypical or absent symptoms of ischemia in DM patients and can be one cause of worse outcomes.

### 4.4. PCI Procedures

Suboptimal results of pPCI with slower flow in the infarct-related artery was more often observed in the diabetic group. The finding is associated with a poor prognosis and it is generally believed to be the result of more prevalent distal embolization enhanced by lower coronary reserve and more diffuse coronary disease in diabetics. This highlights the importance of aggressive antithrombotic drug regimens to manage this population. Both prasugrel and ticagrelor have the potential to reduce distal embolization, but the time from randomization to infarct-related artery reperfusion in PRAGUE-18 was too short to expect full antiplatelet effects from either drug at the time of the procedure. This finding emphasizes the need for treatment of DM STEMI patients as soon as possible. If efficacious pretreatment is not possible, the use of fast-acting antiplatelet agents, such as cangrelor or a GP IIb/IIIa inhibitor, might be considered in DM patients. Patients with DM also have increased levels of procoagulation factors e.g., fibrinogen, tissue, or von Willebrand factors, and decreased levels of anticoagulation factors such as protein C and antithrombin III [18]. Specific more aggressive anticoagulation treatment during pPCI in patients with DM might therefore be effective.

### 4.5. Outcomes

Our findings extend to prior observations on the adverse effect of diabetes on STEMI prognoses. Historically, it has been reported that patients with AMI and DM have a two to a fivefold higher risk for recurrent cardiovascular events, including death, compared to subjects without DM. Registries from the early 2000s report the cardiac mortality of patients with DM and acute coronary syndrome (ACS) are 2- to 3-fold higher compared to patients without DM [19]. The hospital mortality rate of diabetics at that time was 13% [20,21]. Randomized studies from that period showed a 1.6–2-fold higher mortality of diabetic patients after STEMI, with hospital mortality of about 5% and one-year mortality over 13% [16,22]. In the control arm of a meta-analysis testing GP IIb/IIIa inhibitors in pPCI, the mortality rate was four-fold higher in diabetics compared with non-diabetics [23]. A recent Atlantic study reported a 2-fold increase risk of clinically important ischemic events in diabetics [24].

In the PRAGUE-18 study, despite a very high-risk AMI population, a one-year mortality rate of 6.8% among DM patients was low compared to older studies. However, DM remained associated with a 2-fold higher risk of reinfarction. DM patients on insulin were associated with 2.5-fold higher mortality at one year. These findings show an improving prognosis for DM patients, but the relative prognostic impact of DM remains. Diffuse coronary atherosclerosis, reduced coronary reserve, poorer collaterals, reduced compensatory capacity of the myocardium, and diabetic cardiomyopathy in DM patients likely play a role [25].

DM patients use insulin mainly when other treatments are inadequate. The negative impact of insulin treatment on prognosis is likely related to more advanced DM among these patients. However, a direct negative impact of insulin on the prognosis cannot be excluded. Adverse effects of hyperinsulinemia on coagulation and smooth muscle cell proliferation and migration are well known.

### 4.6. Stent Thrombosis

In older studies, DM was associated with a higher risk of stent thrombosis with an up to a 5-fold increase [26]. This was not surprising considering the higher platelet reactivity in DM patients whose platelet response to clopidogrel had attenuated [27]. In the PRAGUE-18 study, the incidence of stent thrombosis in DM patients was 2% (HR 2.377; 95% CI, 0.864–6.541; *p* = 0.084) (Appendix A).

### 4.7. Bleeding

Unlike in older cohorts, we did not observe any increased risk of bleeding in DM patients compared to patients without DM (HR 0.861; 95% CI, 0.554–1.339; *p* = 0.506) (Appendix A). A similar result was seen in the TRITON TIMI 38 study, where DM was associated with higher bleeding risk in the clopidogrel but not in the prasugrel treatment groups. This was particularly interesting since there was a higher bleeding risk associated with prasugrel in patients without DM [12]. Why prasugrel does not increase the bleeding risk in diabetics, while clopidogrel does, is not fully elucidated.

### 4.8. Causes of Higher Risk

One of the main causes of the negative prognostic impact of DM is believed to be increased platelet reactivity, a prothrombotic state, and a higher risk of thrombus embolization, which is poorly tolerated due to the more diffuse coronary disease, lower coronary reserve, and poorer collateralization in diabetics [25]. Furthermore, DM patients were more likely to have a poor response to clopidogrel [27]; the impact of platelet reactivity on cardiovascular events in patients with DM and coronary artery disease is documented [6].

The extraordinary role of platelets in DM was observed in the studies, where the use of glycoprotein IIb/IIIa inhibitor abciximab with stent implantation during pPCI significantly decreased the risk of cardiovascular events in diabetics by one half [23,28].

## 5. Conclusions

Despite modern coronary interventions with new generation drug eluting stenting, intense antithrombotic therapy and high levels of guideline-based medical care, diabetes still has significant adverse effects on the prognosis of STEMI patients, which highlights the importance of further improvements in the management of this high-risk population.

The new ESC guidelines for diabetes management with the use of SGLT2 inhibitors and with the new DM specific targets might mitigate this significant risk factor in patients with STEMI.

## 6. Study Limitations

Although pre-specified, the present study is a subgroup post hoc analysis of the PRAGUE-18 trial with its limitations.

## Figures and Tables

**Figure 1 jcm-09-02555-f001:**
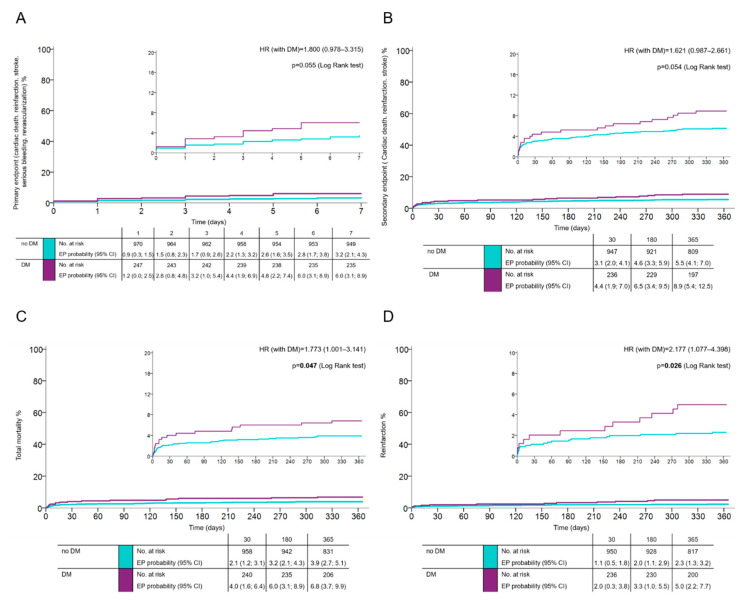
Patients with and without DM following acute myocardial infarction (AMI) treated with primary percutaneous coronary intervention and prasugrel or ticagrelor. The occurrence of (**A**): the primary key efficacy endpoint (cardiovascular death, spontaneous re-infarction, stroke, serious bleeding, and revascularization at 7 days), (**B**): the secondary endpoint (cardiovascular death, spontaneous re-infarction, and stroke at 1 year), (**C**): total mortality at 1 year, and (**D**): reinfarction at 1 year (time-to-event analysis was done using the Kaplan–Meier estimate of the survival function).

**Figure 2 jcm-09-02555-f002:**
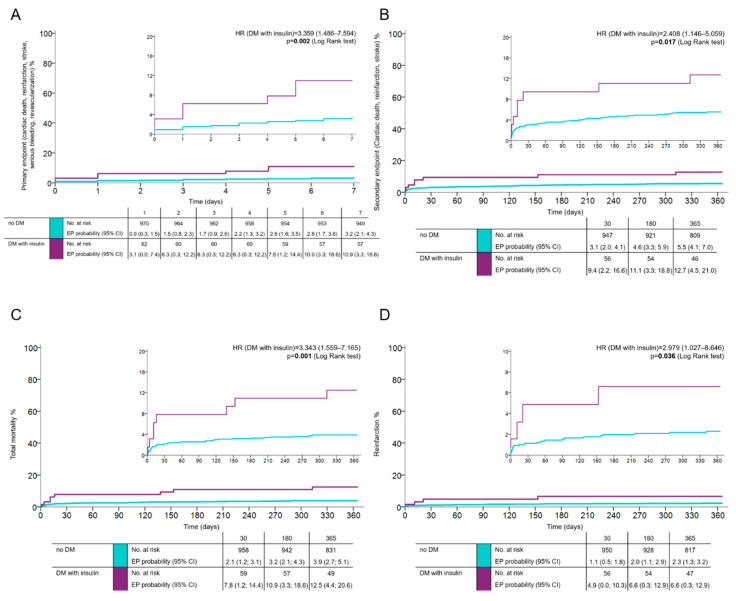
Patients with DM on insulin therapy compared to patients without DM, following acute myocardial infarction (AMI) treated with primary percutaneous coronary intervention and prasugrel or ticagrelor. The occurrence of (**A**): the primary key efficacy endpoint (cardiovascular death, spontaneous re-infarction, stroke, serious bleeding, and revascularization at 7 days), (**B**): the secondary endpoint (cardiovascular death, spontaneous re-infarction, and stroke at 1 year), (**C**): total mortality at 1 year, (**D**): and reinfarction at 1 year (time-to-event analysis was done using the Kaplan–Meier estimate of the survival function).

**Table 1 jcm-09-02555-t001:** Baseline characteristics of the PRAGUE 18 study population relative to diabetes mellitus (DM).

	Without DM (N = 980)	With DM (N = 250)	*p*-Value
Men	764 (78.0%)	167 (66.8%)	<0.001
Age (years)	60.8	65.9	<0.001
Obesity (BMI > 30)	163 (16.6%)	77 (30.8%)	<0.001
Median time from symptom onset to hospital arrival	2.5 h	3.1 h	0.109
**Killip classification**			
I	867 (88.5%)	219 (87.2%)	0.872
II	64 (6.5%)	18 (7.2%)	
III	14 (1.4%)	3 (1.2%)	
IV	35 (3.6%)	11 (4.0%)	
**ECG on admission**			
STEMI	900 (91.8%)	235 (94.0%)	0.290
LBBB	13 (1.3%)	5 (2.0%)	0.387
RBBB	18 (1.8%)	5 (2.0%)	0.797
NSTE MI	56 (5.7%)	10 (4.0%)	0.346
**History**			
Hyperlipidemia	318 (32.4%)	104 (41.6%)	0.007
Hypertension	455 (46.4%)	175 (70.0%)	<0.001
Current smoker	653 (66.6%)	145 (58.0%)	0,012
Previous MI	76 (7.8%)	27 (10.8%)	0.126
Previous PCI	63 (6.4%)	24 (9.6%)	0.096
Previous CABG	16 (1.6%)	5 (2.0%)	0.784
Chronic heart failure	8 (0.8%)	4 (1.6%)	0.278
Chronic kidney disease	12 (1.2%)	4 (1.6%)	0.547
Peripheral artery disease	25 (2.6%)	11 (4.4%)	0.140
Bleeding	2 (0.2%)	4 (1.6%)	0.018
**Chronic therapy before admission**			
Aspirin	126 (12.9%)	66 (26.4%)	<0.001
Beta Blocker	145 (14.8%)	82 (32.8%)	<0.001
ACEi	193 (19.7%)	89 (35.6%)	<0.001
ARB	92 (9.4%)	40 (16.0%)	0.004
Statin	143 (14.6%)	75 (30.0%)	<0.001
Proton Pump inhibitor	50 (5.1%)	26 (10.4%)	0.003
**Initial laboratory evaluation**			
Hemoglobin; Median, g/L	145.0	141.0	<0.001
Platelet count; Median, (× 10^9^/L)	225.0	226.0	0.346
Urea; Median, mmol/L	5.1	6.0	<0.001
Creatinine; Median, umol/L	82.0	82.0	0.434
**Coronary angiography and pPCI**			
Initial TIMI Flow grade < 3	44 (4.5%)	13 (5.2%)	<0.615
Vessel disease > 1	480 (49.0%)	144 (57.8%)	<0.016
Left main—disease	27 (2.8%)	14 (5.6%)	<0.031
**Culprit artery**			
LMCA	6 (0.6%)	6 (2.4%)	0.020
LAD	383 (39.1%)	93 (37.2%)	0.611
LAD (DB)	54 (5.5%)	15 (6.0%)	0.759
Cx	110 (11.2%)	25 (10.0%)	0.651
Cx (MB)	68 (8.2%)	19 (7.6%)	0.680
RCA	401 (40.9%)	112 (44.8%)	0.012
Thrombus aspiration	76 (30.6%)	314 (32.3%)	0.648
Drug eluting stent implantation	163 (68.2%)	624 (66.5%)	0.645
GP IIb/IIIa inhibitor use	62 (24.8%)	186 (19%)	0.043
Procedural failure or suboptimal result of pPCI	39 (4.0%)	20 (8.0%)	0.012
**Discharge therapy**			
Aspirin	961 (98.5%)	245 (98.8%)	0.999
Beta-blocker	797 (81.7%)	213 (85.9)	0.134
ACEi/ARB	819 (83.9%)	212 (85.5%)	0.626
Statin	927 (95%)	231 (93.1%)	0.270
Proton Pump inhibitor	599 (61.4%)	154 (62.1%)	0.884

Absolute and relative frequencies for categorical variables tested using the Fisher exact test; Continuous data described as medians with a 5th to 95th percentile, tested using the Mann–Whitney test. The body-mass index (BMI) = weight in kilograms divided by the square of the height in meters, ECG—electrocardiogram, STEMI—ST-elevation MI, LBBB—left bundle branch block, RBBB—right bundle branch block, NSTEMI—non-ST-elevation MI, CABG—coronary artery bypass grafting, ACEi—angiotensin-converting enzyme inhibitor, ARB—angiotensin receptor inhibitor, LAD—left anterior descending artery, DB—diagonal branch, LMCA—left main coronary artery, RCA—right coronary artery, Cx—circumflex artery, MB—marginal branch, and pPCI—primary percutaneous intervention.

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
