# Peer review of "The Effect of Diabetes on Prognosis Following Myocardial Infarction Treated with Primary Angioplasty and Potent Antiplatelet Therapy"

_jcm, 2020, doi:10.3390/jcm9082555_

Round 1

Reviewer 1 Report

Thank you very much for the possibility to review this manuscript. It is interesting and relevenat. However, english language should be improved. I suggest to revise it by a native speaker. The content of the paper as well as the methods and results seem to be sound. However, it is hard to read and should be revised to be easier to understand.

Author Response

Thank you for your comment, which we fully accept. The paper has been revised and corrected by professional native speaker editor now.

Reviewer 2 Report

Thank you for inviting me to review the article "THE EFFECT OF DIABETES ON PROGNOSIS FOLLOWING MYOCARDIAL INFARCTION TREATED WITH PRIMARY ANGIOPLASTY AND POTENT ANTIPLATELETS" (jcm-849041).
The article was aimed to investigate the impact of DM on the prognosis of patients with acute myocardial infarction. My comments are as follows.
1. The article needs English editing for minor spelling and grammar errors.
2. Table 1 shows the baseline characteristics of enrollees vary quite a lot between the two groups. The authors should perform a multivariate analysis to determine the impact of DM on the endpoints.
3. In Line 159, the authors mentioned patient randomization in their study. Please provide the details of randomization because I did not find randomization in the study.
4. The authors found that Insulin-treated diabetics exhibited all assessed endpoints more frequently when compared with diabetic patients not on insulin. Were patients not in insulin because they had minor DM and did not need insulin control? Or they used OHA? Or they did not even have DM control? The results might hint that insulin was associated with worse outcome in DM patients with AMI. The authors should clearly explain the aim and design of this analysis and the implication of the results as well. 

Author Response

We would like to thank all referees for the comments on this manuscript. We greatly appreciate the input, and we have enclosed a revised version that will hopefully address the remarks and recommendations in the reviews; the changes are addressed individually below.

Author’s replies are in italic letters.

1. The article needs English editing for minor spelling and grammar errors.

Thank you for your comment, which we fully accept. The paper has been revised and corrected by professional native speaker editor now.

2. Table 1 shows the baseline characteristics of enrollees vary quite a lot between the two groups. The authors should perform a multivariate analysis to determine the impact of DM on the endpoints.

Yes, multivariate analysis was done and only age, sex and presence of multivessel disease besides DM showed significant impact on the endpoints.

3. In Line 159, the authors mentioned patient randomization in their study. Please provide the details of randomization because I did not find randomization in the study.

The randomization process was described previously in the main papers of PRAGUE 18 study. This information was added to the section Methods.

The article was corrected as follows:

line 125: Enrollment criteria, the design and randomization process of the PRAGUE-18 study have been previously described [13,14].

Line 117: Simple randomization with GraphPad scientific software was adopted for the study. The sealed envelope method was used for distribution of randomization codes.

4. The authors found that Insulin-treated diabetics exhibited all assessed endpoints more frequently when compared with diabetic patients not on insulin. Were patients not in insulin because they had minor DM and did not need insulin control? Or they used OHA? Or they did not even have DM control? The results might hint that insulin was associated with worse outcome in DM patients with AMI. The authors should clearly explain the aim and design of this analysis and the implication of the results as well. 

Thank you for your comment, which is very relevant. Yes, the insulin control mostly means major DM, where OHA control is not sufficient.

The article was corrected as follows:

Line 111: The impact of DM and impact of DM regarding insulin control on patient prognosis (study endpoints) was evaluated separately.

Line 296: DM patients use insulin, mainly when other treatments are inadequate. The negative impact of insulin treatment on prognosis is likely related to more advanced DM among these patients. However, a direct negative impact of insulin on the prognosis cannot be excluded. Adverse effects of hyperinsulinemia on coagulation and smooth muscle cell proliferation and migration are well known.

Reviewer 3 Report

This is a valid analysis of a relevant question.

I recommend the following improvements before publication:

  • Replace "multivariate" with "multivariable" throughout the manuscript.
  • The first sentence of the abstract "whether...can change the impact of... on..." sounds like the authors would investigate interaction between two risk factors (therapy and DM), which they did not. In fact they analyzed the effect of one risk factor (DM) in a defined group of patients (those with the up to date-therapy). Please reword. (The introduction-section in the main text has a good wording for this)
  • The conclusion of the abstract should highlight the fact that no significant effect was found for primary and secondary endpoints.
  • Multivariable analysis included only 3 variables (age, sex and presence of multivessel disease) besides DM. Please elaborate on how and why those parameters were chosen (Based on clinical relevance? I could think of a number of additional parameters with clinical relevance. Based on univariable testing? This is usually not a good idea. Etc.)
  • I would recommend removal of all the p-values from univariable testing from Table 1, but leave this decision up to the editors.
  • Conclusion: Same applies here as for conclusion of the abstract

Author Response

We would like to thank all referees for the comments on this manuscript. We greatly appreciate the input, and we have enclosed a revised version that will hopefully address the remarks and recommendations in the reviews; the changes are addressed individually below.

Author’s replies are in italic letters.

1. Replace "multivariate" with "multivariable" throughout the manuscript.

Thank you for correction. The "multivariate" was replaced with "multivariable" throughout the whole manuscript.

2. The first sentence of the abstract "whether...can change the impact of... on..." sounds like the authors would investigate interaction between two risk factors (therapy and DM), which they did not. In fact they analyzed the effect of one risk factor (DM) in a defined group of patients (those with the up to date-therapy). Please reword. (The introduction-section in the main text has a good wording for this)

Thank you for your comment.

The article was corrected as follows:

Line 33: The original text was replaced with: “To investigate the prognostic significance of DM in patients with high risk AMI treated with pPCI in the era of potent antithrombotics”.

3. The conclusion of the abstract should highlight the fact that no significant effect was found for primary and secondary endpoints.

Thank you for your comment. “The primary net-clinical endpoint (EP) containing death, spontaneous nonfatal MI, stroke, severe bleeding, and revascularization at day 7 occurred in 6.1% of patients with, and in 3.5% of patients without DM (HR 1.8; 95% CI 0.978–3.315; P = 0.055). At one year the key secondary endpoint defined as cardiovascular death, spontaneous MI, or stroke occurred in 8.8% and 5.5% (HR 1.621; 95% CI 0.987–2.661; P = 0.054) respectively.” The p-values for comparisons of occurrences of primary and secondary end-points were both close to statistical significance. There is a high probability that with larger patient population the differences would reached significance. Therefore we prefer not include statement about differences in combined primary and key secondary endpoints in conclusions. However, if editors decided, that these findings need to be highlighted not only in results section, we will add the information also in the conclusion.

4. Multivariable analysis included only 3 variables (age, sex and presence of multivessel disease) besides DM. Please elaborate on how and why those parameters were chosen (Based on clinical relevance? I could think of a number of additional parameters with clinical relevance. Based on univariable testing? This is usually not a good idea. Etc.)

Thank you for your comment. The selection of variables for multivariable model was based on forward stepwise algorithm applied on set of variables selected by their clinical relevance and / or results of univariate analysis.

5. I would recommend removal of all the p-values from univariable testing from Table 1, but leave this decision up to the editors.

Thank You, yes we will leave this decision up to the editors.

6. Conclusion: Same applies here as for conclusion of the abstract

Thank you for your comment. The same as comment no. 3 applies, please.

Reviewer 4 Report

Ref: jcm-849041
Title: THE EFFECT OF DIABETES ON PROGNOSIS FOLLOWING MYOCARDIAL INFARCTION TREATED WITH PRIMARY ANGIOPLASTY AND POTENT ANTIPLATELETS

The manuscript “THE EFFECT OF DIABETES ON PROGNOSIS FOLLOWING MYOCARDIAL INFARCTION TREATED WITH PRIMARY ANGIOPLASTY AND POTENT ANTIPLATELETS” by Simek et al. aimed to evaluate if the prognostic significance of DM in patients with AMI treated with pPCI in the era of potent antithrombotics and to investigate whether the currently most efficient treatment available – the primary PCI and potent P2Y12 inhibitors can change the negative impact of DM on prognosis of patients with the highest risk AMI.

This is a reasonably large study, including 1230 patients, with standardized outcomes in relation to the impact of DM on outcomes following MI. The statistical analysis are appropriate.

Was there any difference in outcomes between patients with newly diagnosed DM during the hospitalization for MI and patients with established DM?

Was there any difference in outcomes between those with T2DM and T1DM?

Please report medical therapy at discharge in Table 1. 

Author Response

We would like to thank all referees for the comments on this manuscript. We greatly appreciate the input, and we have enclosed a revised version that will hopefully address the remarks and recommendations in the reviews; the changes are addressed individually below.

Author’s replies are in italic letters.

1. Was there any difference in outcomes between patients with newly diagnosed DM during the hospitalization for MI and patients with established DM?

This is a very interesting suggestion for future analyses/investigation. But now we do not have answer from our data.

2. Was there any difference in outcomes between those with T2DM and T1DM?

This again is very interesting question. Unfortunately, we do not have data for this analysis.

3. Please report medical therapy at discharge in Table 1. 

Thank You for relevant comment. The discharge medical therapy was added to Table 1.

Round 2

Reviewer 2 Report

The study design of the revised manuscript did not answer my concern in my previous review report:

The authors found that Insulin-treated diabetics exhibited all assessed endpoints more frequently when compared with diabetic patients not on insulin. Were patients not in insulin because they had minor DM and did not need insulin control? Or they used OHA? Or they did not even have DM control? The results might hint that insulin was associated with worse outcome in DM patients with AMI. The authors should clearly explain the aim and design of this analysis and the implication of the results as well.

Author Response

Thank you for your comment.

We present a standard post-hoc analysis of diabetic population randomized into a cardiovascular outcome study. Presence of DM along with a type of glycemic control was based on patient's history and initial clinical examination. All diabetics on a diet, OHA or insulin control were included. The reason for choosing the type of a pre-randomization glycemic control was not the subject of this analysis. Similarly, for instance we were not evaluating the choice of treatment of the peripheral artery disease (intervention, surgery, drugs, etc...) for the purposes of the analysis of the study outcomes in a specific subgroup population.  Therefore, we think that the relevance of this comment is highly debatable.

The article was corrected as follows:

Line 119: The diagnosis of DM was based on patients’ history and on initial clinical examination. All diabetics on a diet, oral hypoglycemic medication or insulin control were included. Subgroup of patients requiring insulin control was evaluated separately as these patients present mostly more advanced DM and are at higher risk. Nevertheless, the reasons for pre-randomization choice between insulin and other treatments were not analyzed. 

Line 169: one-fourth (N = 64) of the diabetic patients were on long-term insulin treatment, 58.4% on oral hypoglycemic medication, and 16% were controlled by the diet only.

Line 220: Patients with more advanced DM requiring insulin control exhibited all assessed 

Line 304: DM patients use insulin mainly when other treatments are inadequate. The negative impact of insulin treatment on prognosis is likely related to more advanced DM among these patients. However, a direct negative impact of insulin on the prognosis cannot be excluded. Adverse effects of hyperinsulinemia on coagulation and smooth muscle cell proliferation and migration are well known.